# A combined HMM–PCNN model in the contourlet domain for image data compression

**Guoan Yang***, **Junjie Yang, Zhengzhi Lu, Yuhao Wang**

School of Automation Science and Engineering, Xian Jiaotong University, Xi'an, Shaanxi, China

* gayang@mail.xjtu.edu.cn

## Abstract

Multiscale geometric analysis (MGA) is not only characterized by multi-resolution, time-frequency localization, multidirectionality and anisotropy, but also outdoes the limitations of wavelet transform in representing high-dimensional singular data such as edges and contours. Therefore, researchers have been exploring new MGA-based image compression standards rather than the JPEG2000 standard. However, due to the difference in terms of the data structure, redundancy and decorrelation between wavelet and MGA, as well as the complexity of the coding scheme, so far, no definitive researches have been reported on the MGA-based image coding schemes. In addressing this problem, this paper proposes an image data compression approach using the hidden Markov model (HMM)/pulse-coupled neural network (PCNN) model in the contourlet domain. First, a sparse decomposition of an image was performed using a contourlet transform to obtain the coefficients that show the multiscale and multidirectional characteristics. An HMM was then adopted to establish links between coefficients in neighboring subbands of different levels and directions. An Expectation-Maximization (EM) algorithm was also adopted in training the HMM in order to estimate the state probability matrix, which maintains the same structure of the contourlet decomposition coefficients. In addition, each state probability can be classified by the PCNN based on the state probability distribution. Experimental results show that the HMM/PCNN -contourlet model proposed in this paper leads to better compression performance and offer a more flexible encoding scheme.

## 1 Introduction

The ability of multiscale geometric analysis (MGA) theory to process high-dimensional data is better than that of wavelet transform [1]. Since the birth of the JPEG2000 standard, researchers have conducted extensive research on image coding based on the MGA method, among which the most representative research results include image coding based on Ridgelet transform [2–5], Curvelet transform [6, 7], Contourlet transform [8–16], Bandelet transform [17, 18], and based on directional wavelet transform [19], etc. Besides, due to computational complexity and redundancy problems, these researches focused primarily on Contourlet transform. Specifically, in 2004, Eslami and Radha proposed an image coding method, wavelet-based

**Data Availability Statement:** All relevant data are within the paper and its Supporting Information files.

**Funding:** This resubmission is supported by the national natural science foundation of China (Grant

No. 61673314). Website is http://www.nsfc.gov.cn/english/site_1/index.html. Project leader is Guoan Yang.

**Competing interests:** The authors have declared that no competing interests exist.

contourlet transform (WBCT), using CDF 9/7 wavelet instead of Laplacian pyramid (LP) decomposition to solve the 4/3 redundancy in the Contourlet, and also achieved good image compression performance [9, 10]. Nguyen constructed a new directional filter bank (DFB) in 2005 with six high-pass directional subbands and two low-pass directional subbands, and proposed a new contourlet transform, whose image coding performance is better than that of wavelet transform [20]. In 2009, Tanaka [21] proposed a new type of contourlet transform in combination with a two-dimensional DFB bank and directional wavelets, which were both simple implementation and low calculation costs, and its image coding performance is superior to the contourlet proposed by Minh N. Do [1]. In 2012, Hong and Hang built a short-type directional filter, whose DFB is implemented in only a few selected directional subbands. The selection of subbands was accomplished by a mean-shift-based decision procedure, and the embedded subband coding with optimized truncation (ESCOT) coding was adopted, which reduced computational complexity and achieved better image compression performance [22]. In 2013, Gehrke et al. [23] established a mathematical relationship between the coding gain and the DFB coefficient, and a new DFB was proposed based on the numerical optimization criterion of DFB coding gain in the lifting scheme, and its image compression performance outperformed the contourlet. In 2015, Naimi and Beloulata [24] proposed a multiple description image coding method based on contourlet transform, which can reliably transmit useful information to the encoder, so as to effectively code when packet loss occurs during the transmission of compressed code stream. They also demonstrated that the image coding performance surpassed wavelet transform. In 2016, Nejati et al. [25] presented the boosted multi-scale dictionary learning in the wavelet domain for image compression, which had better image compression performance than JPEG, JPEG2000 and JPEG-XR methods. Besides, there is also a need to clarify that MGA-based coding can be utilized not only for image data compression, but also for digital watermarking [26], dictionary learning [27, 28], image quality assessment [29], image recognition [30, 31], and so on.

In recent years, research on contourlet based image coding tends to be combined with human visual features. This is because the natural image itself contains a lot of redundant information, which can be further compressed [32, 33]; simple cells in the visual cortex receptive field possesses a sparse coding mechanism, which can remove a lot of redundant information and capture only useful information [34–36]. This is selective attention to visual perception [37–39]. The idea of considering human visual features for MGA can best be realized in image data compression applications. For this reason, we would like to report the result of research on image data compression of combined HMM/PCNN model in the contourlet domain, which is based on the previous research [40].

The paper is organized as follows: In Section 2, the study introduces some theoretical preparations, including the contourlet transform and the HMM-contourlet model. In Section 3, a PCNN model and an adaptive PCNN are illustrated. Section 4 gives the SPIHT algorithm and its procedure. Section 5 explains how to adjust the SPIHT algorithm in the contourlet domain and combine the HMM model with a PCNN to realize a hybrid HMM–PCNN, thereby improving the SPIHT algorithm by classifying the coefficients. Section 6 describes the algorithm used in the proposed approach, the experimental results and their analysis, and conclusions are discussed in Section 7.

## 2 Theoretical preparation

### 2.1 Contourlet transform

The contourlet transform is composed of a Laplacian pyramid (LP) and directional filter banks (DFBs), where the former composition extracts the high-frequency partition of an image,

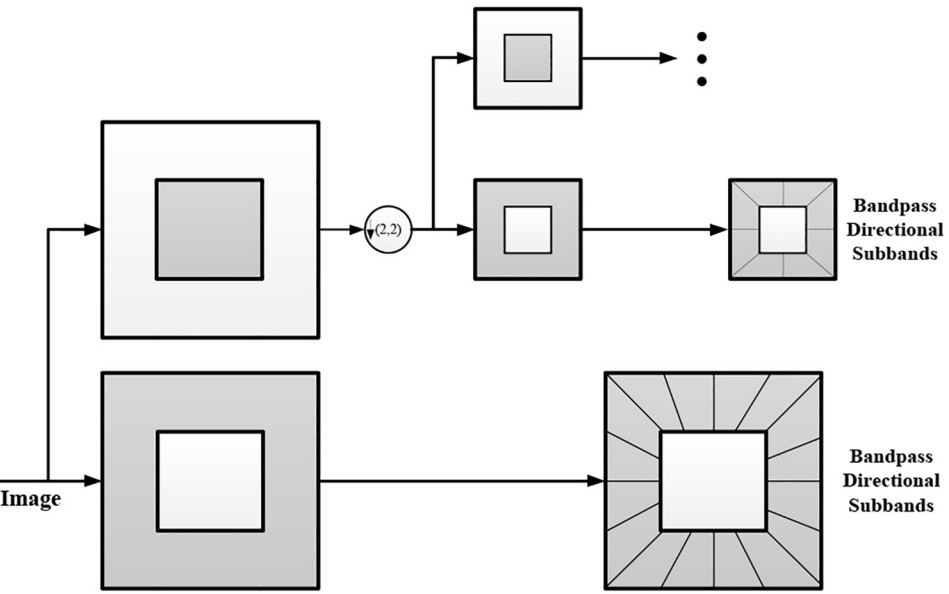

**Fig 1. Iterated LP decomposition from coarse signal to fine signal.**

while the latter composition collects the directional information of the high-frequency partition and achieves a more efficient performance in terms of sparse representation [1]. Contourlet transform usually first uses multiscale decomposition to capture the point discontinuities distributed along the geometric edges of an image, which is similar to that of wavelet, and then the transform links the closed point discontinuities into linear structures, namely contour segments, based on the directional information. This process can be iterated from a coarse signal to a detailed signal, resulting in a series of bandpass images, as shown in Fig 1.

In 2005, Do and Vetterli [1] proposed a simplified DFB, which is intuitively constructed by two blocks. The first block is a quincunx filter bank which has two channels, which can divide a 2-D spectrum plane into horizontal and vertical portions. The second block is a shearing operator adopted for reordering the samples of the image, as is shown in Fig 2.

An LP and DFBs are combined to form a two-stage filter bank in the contourlet transform. DFBs are used for capturing the direction information of the high-pass partition. However, some low-pass partitions may leak into the high-pass partitions, implying that a simple DFB cannot achieve the sparse representation of an image. Therefore, an LP is used to make up for this drawback of DFB: after removing the low-pass partition of an image, the DFB processes the high-pass partition and links the point discontinuities to describe the geometric edges of the image, namely, the contours.

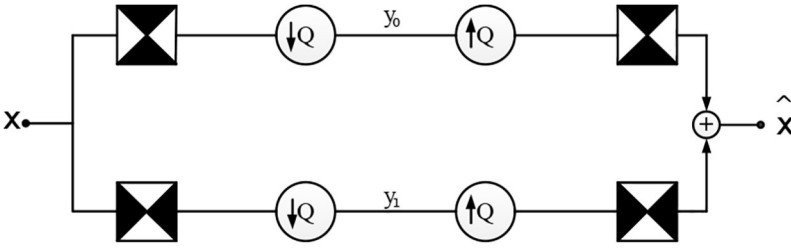

**Fig 2. A 2-D spectrum partitioning using quincunx filter banks with fan filters.**

## 2.2 HMM in contourlet domain

An image can be decomposed into many coefficients distributed in many directional subbands at different scales. In other words, a subband contains a bound of coefficients that exhibit specific directional features. According to the statistics carried out by Po and Do [41], these coefficients are in generalized Gaussian distribution but marginally non-Gaussian. The entire distribution can be described as a zero-mean Gaussian mixture model, in which the coefficients can be divided into large ones that possess larger variance and present the edges of the geometry, and small ones that possess smaller variance and present the plane areas. A single contourlet coefficient is in one of the two states. Therefore, Po and Do [41] used various hidden states to label the coefficients and link them with the hidden states. There are links between the coefficient and its parent, its cousins, and its neighbors. Statistics show that the parent coefficient is the most significant predictor when considering the three types of related coefficients individually [41]. To reduce the complexity of the model, we only adopt the relationship between the coefficient and its parent. As mentioned above, these coefficients are actually correlated with each other through hidden states. Thus, we adopt an HMM to describe the statistical model. An HMM in contourlet will cause the tree structure to be consistent with the spatial orientation tree in an SPIHT algorithm [15, 42].

In an HMM in the contourlet domain, the hidden state chain is used to describe the state of the coefficient, while the state observation chain is used to describe the value of the coefficient. A parent state is correlated with four child states. From the coarsest scale, the initial hidden state spreads through the tree structure by iteratively multiplying the state transition matrix. The coefficient value can then be calculated by multiplying the hidden state by the observation probability matrix. The tree structure of the HMM is shown in Fig 3. In order to estimate the parameters of the HMM, we used the EM algorithm [43].

## 3 PCNN model

### 3.1 The parametric model of PCNN

Eckhorn [44] proposed a connection model in 1990 that revealed pulse synchronization emissions in the cat's visual cortex. After that, Johnson presented a pulse coupled neural network (PCNN) based on Eckhorn's model in 1999 [45], which implemented a function similar to the visual cortex of the mammalian brain. Different from other artificial neural networks, the PCNN consists of only a single layer which is formed by a 2-D array of neurons without

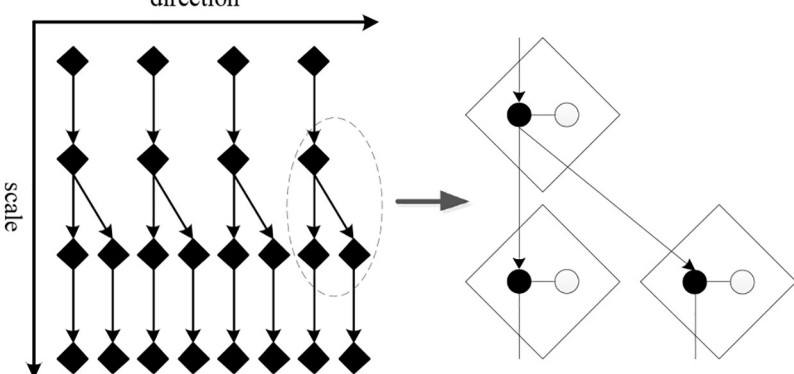

**Fig 3. Links of the subbands of a contourlet decomposed by 4,4,8,8 directions at each scale.** The hidden states (black circles) are correlated to predict the observation states (white circles).

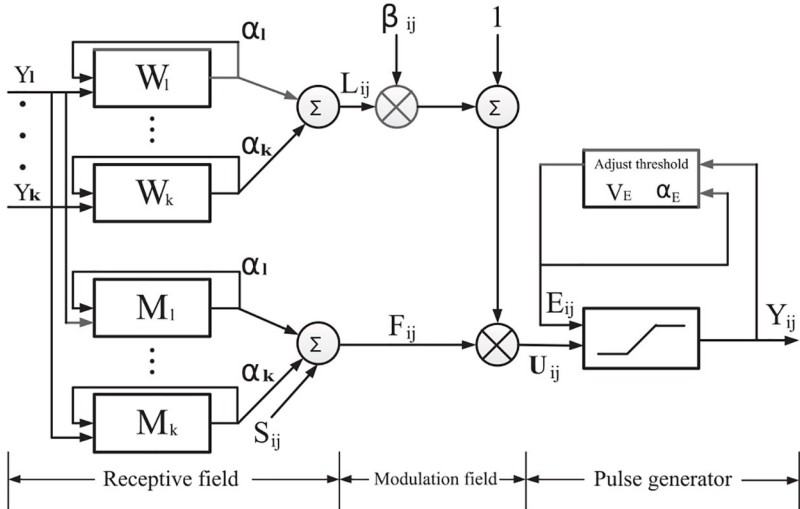

**Fig 4. Structure of a PCNN.**

training. Fig 4 shows a simplified PCNN model, which consists of three parts: the input module, the nonlinear modulation module and the pulse generator. The formulas for describing the PCNN model are as follows:

$$F_{ij}(n) = exp(-\alpha_{kl}^F)F_{ij}(n-1) + S_{ij} + V_F \sum_{kl} M_{ijkl}Y_{kl}(n-1) \tag{1}$$

$$L_{ij}(n) = exp(-\alpha_{kl}^L)L_{ij}(n-1) + V_L \sum_{kl} W_{ijkl}Y_{kl}(n-1) \tag{2}$$

$$U_{ij}(n) = F_{ij}(n)(1 + \beta L_{ij}(n)) \tag{3}$$

$$E_{ij}(n) = exp(-\alpha_E)E_{ij}(n-1) + V_E Y_{ij}(n) \tag{4}$$

$$Y_{ij}(n) = \begin{cases} 1, & U_{ij}(n) \geq E_{ij}(n) \\ 0, & otherwise \end{cases} \tag{5}$$

where $\alpha_{kl}^F$, $\alpha_{kl}^L$ and $\alpha_E$ are the time constants; $V_F$, $V_L$, $V_E$ are the magnitude thresholds; $\beta$ is the linking strength of PCNN. Each neuron is denoted with indices $(i, j)$, and one of its neighboring neurons is denoted as $(k, l)$. $S_{ij}$ is the stimulation of the neuron at $(i, j)$. In $n$th iteration, the feeding input $F_{ij}(n)$ is combined with the linking input $L_{ij}(n)$ to form the internal activity $U_{ij}(n)$ of the neuron. The neuron receives input signals via feeding matrix $M_{ijkl}$, and each neuron is linked to its neighbor, so that the output signal $Y_{ij}(n)$ of a neuron modulates its neighbor's activity via the linking matrix $W_{ijkl}$. Once a neuron is excited, it begins to communicate with its neighbors and encourages it through interconnections of $W_{ijkl}$. When the internal activity $U_{ij}(n)$ is greater than the dynamic threshold $E_{ij}(n)$, the corresponding neuron will be triggered, otherwise the neuron will remain in its earlier state. The internal activity consists of the feeding input and the linking input, so its value is affected by both $F_{ij}$ and $L_{ij}$. Thus, if a neuron is activated, the neighboring neurons with similar tension will also be activated in the next iteration.

Due to its unique structure, PCNN has many outstanding characteristics that are beneficial to image compression and image classification [46]. These characteristics are pulse coupling, nonlinear multiplicative modulation, neighbor-capturing and threshold mechanism of exponential attenuation. Among these characteristics, neighbor-capturing is the most important, because the classification can be implemented by this feature, and an adaptive method designed to obtain a more reasonable $\beta$.

## 3.2 Adaptive PCNN model

Is has been studied that the linking strength $\beta$ has an influence on the activation increment of a neuron. Specifically, larger $\beta$ results in easier activation. Besides, $\beta$ varies with the differences between a certain pixel and its surrounding pixels, which means that larger differences lead to easier activation. Therefore, to establish appropriate $\beta$ values for all neurons, a contrast operator $dec(i, j)$ is adopted in this paper to measure the differences between the neighboring central pixels in the grayscale image.

$$dev(i,j) = \frac{maxf(i,j) - minf(i,j)}{maxf(i,j)}, \beta = dev(i,j) \qquad (6)$$

where $maxf(i, j)$ and $minf(i, j)$ represent the maximum value and minimum value of the contourlet coefficients in the neighborhood of a target pixel. Through this operator, an adaptive $\beta$ can be obtained: high contrast of the local area around a pixel means the corresponding neuron is in an active state and will be easily triggered. Therefore, the local contrast of the target pixel can be normalized to calculate the adaptive $\beta$ as a real parameter of PCNN, and the calculation is expressed as shown in Eq (7), where $S$ is a decomposition subband and $G$ is the gradient of a local area.

$$G_i = \begin{bmatrix} -1 & 0 & 1 \\ -2 & 0 & 2 \\ -1 & 0 & 1 \end{bmatrix} * S, G_j = \begin{bmatrix} 1 & 2 & 1 \\ 0 & 0 & 0 \\ -1 & -2 & -1 \end{bmatrix} * S, G = \sqrt{G_x^2 + G_y^2} \qquad (7)$$

Simialr to the configuration in [40], we set $V_E$ and $\alpha_E$ are set as $V_E = 1 - G$, $\alpha_E = G$ for the adaptive PCNN model.

## 4 SPIHT algorithm

In the SPIHT algorithm [47], the spatial orientation tree is defined as a structure that links the coefficients across adjacent subbands, as shown in Fig 5.

For a certain wavelet coefficient, where $(i, j)$ are the coordinates of the coefficient, four sets are defined as follows: $O(i, j)$ is the set of coordinates of all offspring of the coefficient; $D(i, j)$ is the set of coordinates of all descendants of the coefficient; $H$ is the set of coordinates of all spatial orientation tree roots; $L(i, j) = D(i, j)—O(i, j)$.

SPIHT adopts a classification criterion in which all coefficients are assumed to be either significant or insignificant with respect to established thresholds. That is to say, a coefficient is either important, if it is larger than the threshold, or unimportant, as shown in the function:

$$S_n(T) = \begin{cases} 1, & max(|c_{i,j}|) \geq 2^n \\ 0, & max(|c_{i,j}|) < 2^n \end{cases} \qquad (8)$$

where $T$ is a set of coordinates, $2^n$ is the threshold and $c_{i,j}$ indicates the wavelet coefficients

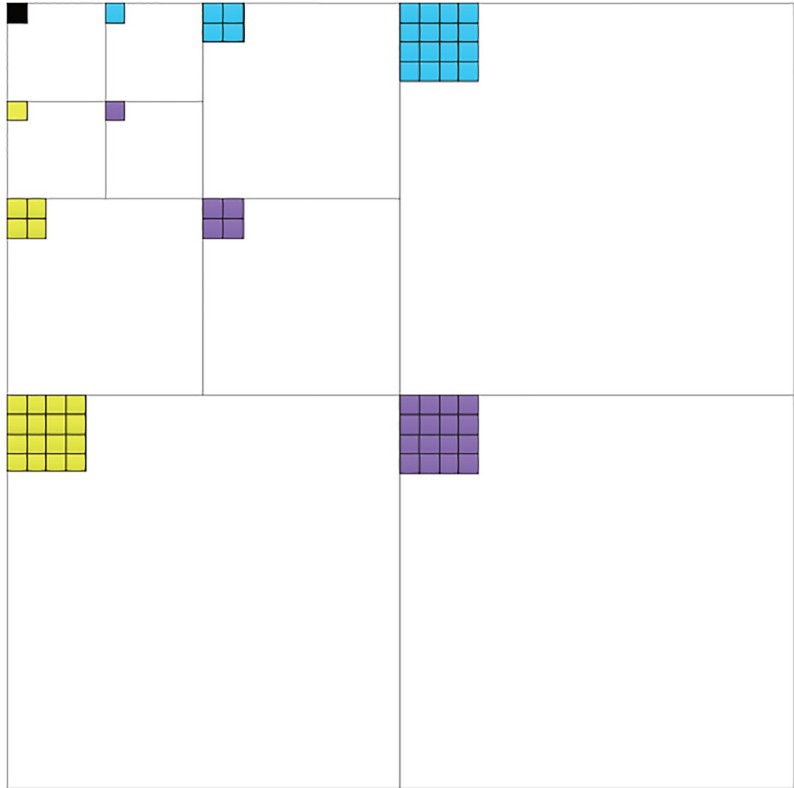

**Fig 5. A spatial orientation tree in a wavelet.** The coefficient colored black has no descendants; other coefficients of same colors possess branches that extend in one direction.

distributed in each subband, indexed by the coordinate subscripts $i$ and $j$. Based on the distribution characteristics of the wavelet coefficients and the structure of the spatial orientation tree, we define three lists to organize the coefficients in implementing the coding: significant pixels (*LSP*), insignificant pixels (*LIP*), and insignificant sets (*LIS*). With these denotations, SPIHT algorithm can be described as follows:

**Initialization** $N = floor(log_2(max(|c(i, j)|)))$, where $N$ indicates the upper limit of the encoding loop; Initialize the threshold $T_0 = 2^N$; Initialize the lists, letting $LIS = \{D(i, j)|(i, j) \in H, chasnonzerodescendant\}$ and label all elements in *LIS* with "A", $LSP = \emptyset$, $LIP = \{(i, j)|(i, j) \in H\}$.

**Sorting pass**

1. For each element $c(i, j)$ in the *LIP*, if $c(i, j)$ is significant, output $S_n(i, j) = 1$. Then move the coordinate of $c(i, j)$ into the *LSP* and output the sign of $c(i, j)$. If $c(i, j)$ is insignificant, then output $S_n(i, j) = 0$.

2. For each element $D(i, j)$ in the *LIS*.

(i). If the elements in $D(i, j)$ are significant and labelled with "A", output $S_n(i, j) = 1$. If the offspring of $D(i, j)$ is significant, then output both the $S_n(i, j) = 1$ and the sign bit, while simultaneously adding the coordinates of the offspring to the *LSP*. If the offspring of $D(i, j)$ is insignificant, output $S_n(i, j) = 0$ and add the coordinates of the offspring to the *LIP*. If the offspring have descendants, move the coordinates of those descendants to the *LIS* and label the corresponding coefficients with "B".

(ii). If the elements in $D(i, j)$ are insignificant, output $S_n(i, j) = 0$. For the pixel set labelled with "B", if the pixel set is significant, output $S_n(i, j) = 1$, add each descendant to the *LIS*, label the coefficient with "A", then remove the descendants from the *LIS*. If the elements in $D(i, j)$ are insignificant, then output $S_n(i, j) = 0$.

**Refinement pass** Output $N$ bits of the absolute value of the coefficient in the last coding level from the beginning, where $N$ is the exponential of the current threshold.

**Quantization scale update** Decrease $N$ by 1 and go to the sorting pass.

## 5 HMM/PCNN-contourlet coding using SPIHT

### 5.1 The method

In [41] Po and Do proposed a contourlet-HMM method that there are excellent performances for image denoising and retrieval. We know further that hybrid HMM/ANN model has been successfully applied in the field of speech recognition. That is because the combined model takes advantage of the pattern classification ability of ANN and the modeling ability of HMM in spatiotemporal use. Therefore, we would try to apply a hybrid HMM/ANN model in the contourlet domain for image processing applications such as image compression. As the third generation of ANN, PCNN model simulates the activity of neurons in cat's visual cortex that not only has good biological characteristics, does not need any training and learning, but also is easy to combine with other methods. Simultaneously PCNN model also indicates the excellent performance in the field of image fusion application. Therefore, in this paper the PCNN model is selected as an ANN model, eventually we propose a hybrid HMM/PCNN model in the contourlet domain for image compression. In brief, The HMM/PCNN model make full use of the HMM's advantages of learning ability, decoding ability, ability to process the time sequence signals, and PCNN's static classification ability. There are several ways to combine the HMM and a PCNN:

1. Normalize the time sequence signal using the HMM, and then input the processed signal into the PCNN to do the classification.

2. Use PCNN to calculate the observation matrix of HMM.

3. Use PCNN to implement the three algorithms (forward algorithm, backward algorithm, Viterbi algorithm) in the HMM.

4. Establish an HMM network.

Therefore, based on all the contents mentioned before, we would like to combine HMM and PCNN to optimize the Po and Do's model, then apply this model to SPIHT to validate its effectiveness.

According to Po and Do's work, the distribution of the contourlet coefficients has the following characteristics [41]:

1. The coefficients in the same subband are distributed as a zero-mean Gaussian mixture conditioned on their parent coefficients.

2. There are two types of coefficients within the Gaussian mixture distribution: the large ones possess a large variance and a low peak, while the small ones possess a small variance and a high peak. The former presents the edges and the latter presents the plane areas.

Based on the distribution characteristics, Po and Do successfully established an HMM contourlet domain, where the state transition matrix and state probability matrix were calculated and used. In order to combine the HMM and PCNN, we need to add PCNN to the HMM in

contourlet. Inspired by the first kind of combination of HMM and PCNN mentioned above, we hope to find some features in the contourlet coefficients for PCNN to process, and the feature space found is the state probability.

The reasons for choosing state probability as classification features are as follows:

1. the magnitudes of the coefficients are unreliable for judging coefficients being large or small, since the magnitude of a certain coefficient may be quite small, but it exists in a Gaussian distribution with large variance;

2. Some coefficients may have near or equal probabilities of being large or small, which may lead to uncertainty in the processing results, because these coefficients cannot be well identified;

3. the EM algorithm for the HMM when describing the same image can result in different parameters, where once the difference between the new parameter and the last parameter is less than the given threshold, the iterative process stops, which causes the process to converge to different parameter sets, leading to the uncertainty of the process result. Therefore, in order to improve the processing efficiency, we use PCNN to produce a Boolean output: to transfer the uncertainty of some coefficients' probabilities into certainty.

Therefore, to implement the classification of the state probability matrix, we adopt the PCNN mentioned above. The state probability matrix has the same multilayer and multidirectional structure as the contourlet subbands. For a certain coefficient $c_{(i, j, k), x}$(denoting the $x$-th coefficient of the $k$th subband in the $j$th direction at the $i$th scale) and its corresponding two-state probabilities $s^l_{(i,j,k),x}$ and $s^s_{(i,j,k),x}$(denoting the large state and small state, respectively), they satisfy

$$p(s^l_{(i,j,k),x}|c_{(i,j,k),x}) + p(s^s_{(i,j,k),x}|c_{(i,j,k),x}) = 1 \qquad (9)$$

The state probabilities distributed in each subband are considered as "pixels" whose gray levels vary between 0 and 1. If the probabilities in the PCNN are inputted with given parameters, the PCNN produces a two-value output, where 0 represents a small state and 1 represents a large state. With this output, a clearer subband of the state probability matrix can be obtained using the Zelda test image, as shown in Fig 6.

To sum up, we first decompose the image into contourlet coefficients, then we establish an HMM in contouret domain and obtain the state transition matrix and state probability matrix. After that, we adopt PCNN to process the state probability matrix to separate the coefficients in the same subband into two groups. Finally, we encode the coefficients with SPIHT.

Since the classification operation of coefficients is considered in our coding scheme, there is need to adjust the original coding method, just as the two classified coefficients are encoded with different compression rates and transmitted separately. After decoding, the two types of classified coefficients are recombined for reconstruction.

## 5.2 Implementation

The flowchart of the HMM/PCNN-contourlet model using the SPIHT algorithm is as follows: firstly, the contourlet transform was adopted for an image to obtain the coefficients, and then HMM was used to model the coefficients in a tree strcuture. HMM underwent training by the EM algorithm to obtain the state probability matrices. Moreover, PCNN was applied to classify the subband coefficients into two groups according to state probability values. Finally, all the subband coefficients were coded, transmitted, and decoded with the SPIHT algorithm.

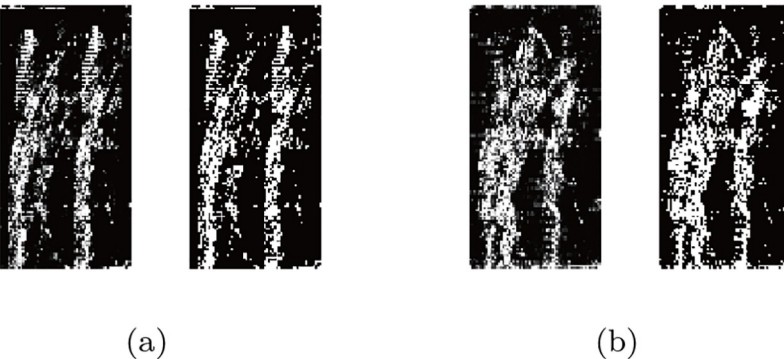

(a)                                    (b)

**Fig 6. Two subbands of the state probability of "Zelda", the leftside of each shows unprocessed subband and the rightside shows processed subband.** It can be observed that the state probability processed by PCNN is clearer than the unprocessed probability and easier to classify.

To implement the SPIHT algorithm in the contourlet, the distribution format of the coefficients must be considered. The distribution of the related contourlet coefficients is shown in Fig 6. Its structure is similar to that of wavelet coding, in which coefficients are distributed in subbands of different scales and different directions. There are also interscale links between coefficients of different scales in adjacent subbands. However, compared to wavelet, two differences still require attention in contourlet coding. The first is that each parent contourlet coefficient has four children, while each parent wavelet coefficient has only three. The second is that the structure of links between related coefficients changes with the variation of decomposition direction at different scales, while stationary links exist among parent wavelet coefficient and their children. Therefore, to implement this coding in the contourlet domain, it is necessary to adjust the spatial orientation tree. The concrete implementation is presented as shown in Fig 7

For the first difference, we established one more directional link for the root coefficients to ensure that, except for the coefficients in the highest frequency partition, each coefficient has four children in the contourlet domain, which makes the original spatial orientation tree

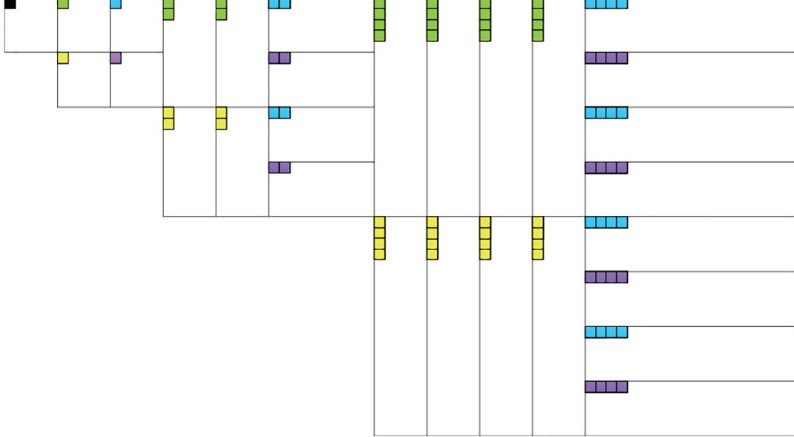

**Fig 7. Related coefficients distribution in a contourlet.** The coefficient colored black is the root of the other colored ones; coefficients of same colors are in the same direction.

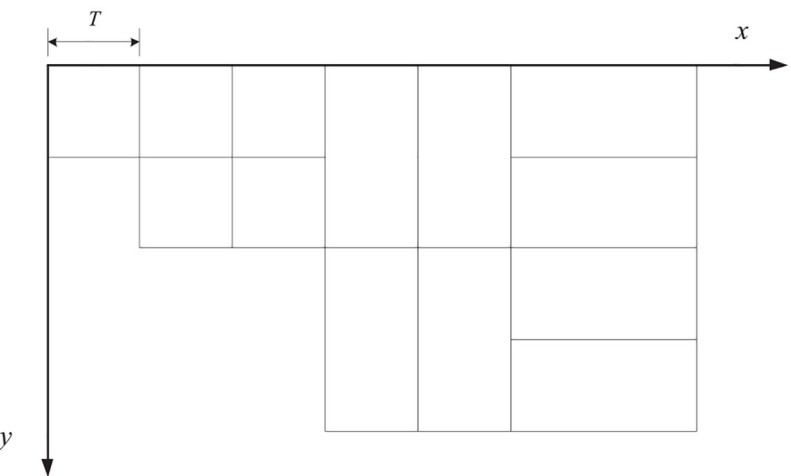

**Fig 8. Coordinates of the whole decomposition distribution.**

become a new tree structure. For the second difference, the link structures between related coefficients on adjacent scales changed when the number of decomposition directions differed on the two scales due to multiple directions in the contourlet. Thus, we improve the indexing method of correlation coefficients in the spatial orientation tree.

In a contourlet transform, each coefficient is correlated with a corresponding node in the tree, and each parent node is related to four children nodes. The process of indexing a specific coefficient is actually realized by searching the coordinates of the nodes in the tree. The decomposition results in various coefficient matrices (or subband matrices) at different levels and directions, but the coefficient matrices in each subband can be processed as a whole, as shown in Fig 8

For a clearer description, assuming that the coordinates of a certain node are $(x, y)$ and the length of the root matrix is $T$; then all possible indexing methods for offspring can be described as follows:

If $(x, y)$ is in the root, then the coordinates of its offspring are

$$(T + x, y) \tag{10}$$

$$(T + x, T + y) \tag{11}$$

$$(2T + x, y) \tag{12}$$

$$(2T + x, T + y) \tag{13}$$

If the $2^{(n+1)}$th subband is the horizontal decomposition, then the coordinates of the offspring are

$$(T + 2x - 1, y) \tag{14}$$

$$(T + 2x, T + y) \tag{15}$$

$$(T + 2x - 1, 2^{n-1}T + y) \tag{16}$$

$$(T + 2x, 2^{n-1}T + y) \tag{17}$$

If the $2^{(n+1)}$th subband is the vertical decomposition, then the coordinates of the offspring are

$$(2^n T + x, 2y - 1) \tag{18}$$

$$(2^n T + x, 2y) \tag{19}$$

$$(2^{n+1} T + x, 2y - 1) \tag{20}$$

$$(2^{n+1} T + x, 2y) \tag{21}$$

We replaced the original coefficient indexing method with this improved one to adapt to the variable spatial orientation tree structure of the contourlet. With this adjustment, a novel SPIHT coding algorithm based on contourlet is proposed. The entire coding procedure is consistent with that of wavelet; the difference lies only in the concrete implementation, as analyzed above. Fig 9 shows the coding results using a Goldhill test image.

In fact, due to the redundancy caused by an LP of the contourlet transform, the application of SPIHT in the contourlet alone cannot achieve better coding performance than SPIHT in wavelets. In [10], they made improvements to solve the problem of redundancy and dependency in the wavelet domain. In their work, LP decomposition was replaced by wavelet decomposition, which maintained a non-redundant transform, as shown in Fig 10. In our study, a hybrid HMM–PCNN was adopted to meet the expectation of a better coding result than a SPIHT contourlet.

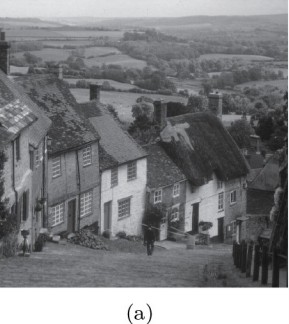 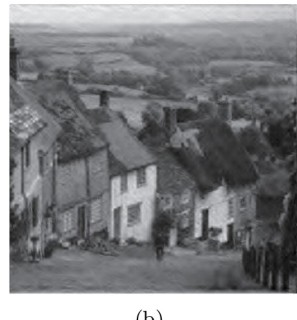 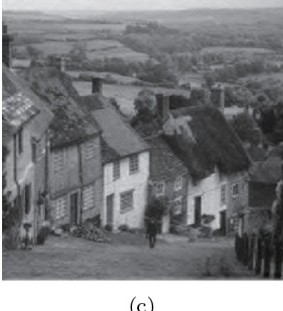

(a)                         (b)                         (c)

**Fig 9.** SPIHT based on a contourlet with 4,4,8,8 directions at each level; (a) Original image, (b) Reconstructed image after coding with 0.15 bpp, (c) Reconstructed image after coding with 0.3 bpp. It can be observed that (c) is clearer than (b) Reprinted from [40] under a CC BY license, with permission from IEEE publisher, original copyright 2019.

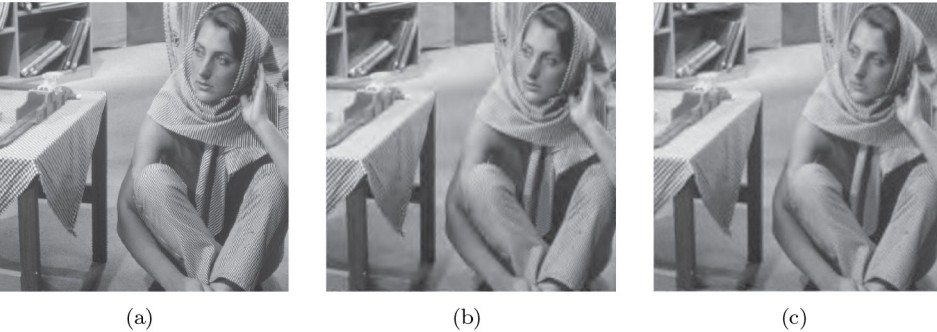

(a)                                    (b)                                    (c)

**Fig 10. Improved contourlet-based SPIHT and contourlet-based SPIHT perfomance.** (a) shows the original image. (b) shows the method in [9], which adopts the contourlet that replaces the LP decomposition with wavelet. (c) shows the image coded with an contourlet based SPIHT. From the images, it can be observed that textures in (b) are more clear than the textures in (c) Reprinted from [40] under a CC BY license, with permission from IEEE publisher, original copyright 2019.

# 6 Experimental results and analysis

## 6.1 PCNN parameters

Based on the forementioned methods for PCNN, we've illustrated some key points to achieve the adaptivity. Herein, other parameters of PCNN are set as follows: $\alpha_{kl}^L = \alpha_{kl}^F = 0.8$; $V_L = V_F = 0.5$; $W_{ijkl} = M_{ijkl} = [1\ 1\ 1;1\ 0\ 1;1\ 1\ 1]$; $N = 10$; where $N$ is the iteration times, $W_{ijkl}$ is the weight matrix of the linking channel connections, and $M_{ijkl}$ is the weight matrix of the feeding channel connections.

## 6.2 Experimental procedure

The entire experimental flowchart is shown in Fig 11.

The algorithm of the contourlet-HMM–PCNN model for SPIHT coding is as follows:

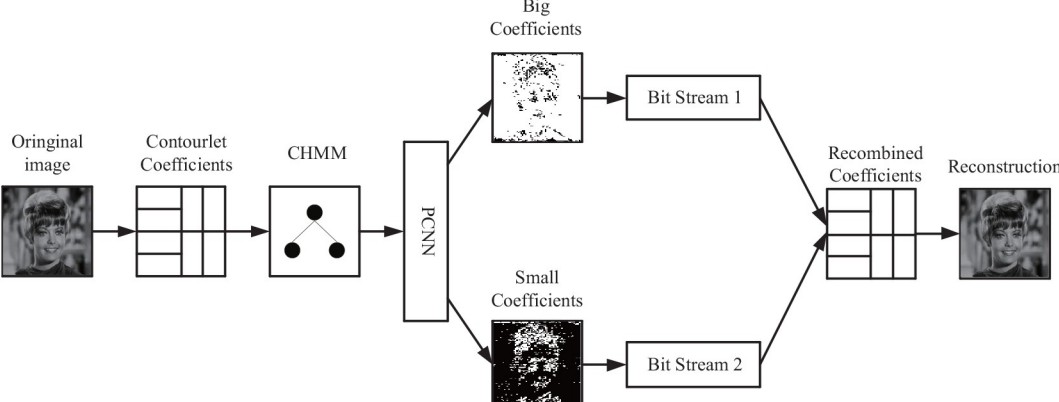

**Fig 11. Hybrid HMM–PCNN model in the contourlet domain for the SPIHT algorithm, where CHMM denotes the HMM in the contourlet domain.** Reprinted from [40] under a CC BY license, with permission from IEEE publisher, original copyright 2019.

1. Use contourlet to decompose the original image into coefficients. The levels of decomposition chosen were [2, 2, 3, 3], which resulted in 4, 4, 8, 8 subbands at each decomposition level. In the contourlet transform, a 9/7 filter was used as a pyramid filter and a PKVA filter was used as a directional directional filter.

2. Train the coefficients with EM algorithm to obtain the HMM in the contourlet domain. With the HMM, the state probability matrix, the Gaussian standard deviation matrix, and the transition probability matrix can be obtained.

3. Input each state probability matrix into the adaptive PCNN for the state probabilities to be classified, and classify the coefficients in the corresponding subbands according to the result. In this step, the PCNN processes the state probabilities and divides them into different groups, which is similar to the segmentation of images in the pixel domain:

    1. Denote a subband partition of the state probability matrix as $s_{j,k,\ n}$.

    2. Set $F_{ij} = 0$, $L_{ij} = 0$, $U_{ij} = 0$, $E_{ij} = 0$, $Y_{ij} = 0$.

    3. Use the $3 \times 3$ matrix of the PCNN linking channel of the PCNN to perform a convolutional operation with the subband.

    4. Calculate $F_{ij}$, $L_{ij}$, $U_{ij}$, $E_{ij}$, $Y_{ij}$. If the maximum iteration number is reached, stop triggering.

    5. Choose an element in the state probability matrix that represents a large coefficient with an output of 1.

    6. Deduct the triggered subband from the original subband to obtain two separated subbands $s_{j,k,n}^{l}$ and $s_{j,k,n}^{s}$.

    7. Multiply the state probability subbands $s_{j,k,n}^{s}$ and $s_{j,k,n}^{l}$ by the corresponding coefficient subbands to obtain the classified coefficients $c_{j,k,n}^{s}$ and $c_{j,k,n}^{s}$.

4. Encode the two groups of classified coefficients using the SPIHT algorithm. Note that different compression ratios can be used for both parts. In our experiment, both groups used the same compression ratios.

5. Receive two bit streams, and combine them into the entire distribution of decoded coefficients, and then recompose the coefficients into a reconstructed image.

## 6.3 Experimental results

This experiment was performed on MATLAB R2018b on a PC with Intel Core i7-7700/3.6 GHz/16 GB. The size of all gray images are 512 rows and 512 columns. To evaluate the performance of the proposed algorithm, we used the comparing criteria like peak signal-to-noise ratio (PSNR) and structural similarity (SSIM). In Table 1, standard SPIHT in wavelet was used as the baseline to make a comparison with the contourlet SPIHT. The abbreviation WT refers to wavelet transform and CT refers to contourlet transform. In Table 2, the large ratio refers to the compression rate of the coefficients with larger variance, and the small ratio refers to the compression rate of the coefficients with smaller variance.

Table 1 shows the image compression results of the SPIHT based on the contourlet transform, while Table 2 shows the image compression results of the SPIHT based on the contourlet-HMM–PCNN model. Both tables use the same image to show the performance of the

**Table 1. Image compression performance of SPIHT.**

| image | Butterfly | | | | Barbara | | | |
|---|---|---|---|---|---|---|---|---|
| Rate(BPP) | 0.1500 | 0.2000 | 0.2500 | 0.3000 | 0.1500 | 0.2000 | 0.2500 | 0.3000 |
| WT PSNR(dB) | **27.645** | **28.203** | **28.543** | **30.125** | **24.563** | **24.998** | **25.296** | **25.677** |
| CT PSNR(dB) | 25.947 | 26.80 | 27.130 | 27.296 | 23.023 | 23.585 | 23.843 | 24.010 |
| WT SSIM | **0.7064** | **0.7592** | **0.8064** | **0.8130** | **0.6531** | **0.6730** | **0.7150** | **0.7624** |
| CT SSIM | 0.6911 | 0.7178 | 0.7408 | 0.7456 | 0.6004 | 0.6329 | 0.6359 | 0.6581 |
| image | Zelda | | | | Goldhill | | | |
| Rate(BPP) | 0.1500 | 0.2000 | 0.2500 | 0.3000 | 0.1500 | 0.2000 | 0.2500 | 0.3000 |
| WT PSNR(dB) | **30.962** | **31.586** | **32.792** | **33.112** | **27.684** | **28.174** | **28.746** | **30.543** |
| CT PSNR(dB) | 30.867 | 30.907 | 32.021 | 32.337 | 25.827 | 26.340 | 26.866 | 27.242 |
| WT SSIM | **0.8306** | **0.8522** | **0.8726** | **0.8591** | **0.7138** | **0.7372** | **0.7862** | **0.8111** |
| CT SSIM | 0.8239 | 0.8247 | 0.8524 | 0.8591 | 0.6109 | 0.6472 | 0.6750 | 0.6944 |
| image | Man | | | | Mandrill | | | |
| Rate(BPP) | 0.1500 | 0.2000 | 0.2500 | 0.3000 | 0.1500 | 0.2000 | 0.2500 | 0.3000 |
| WT PSNR(dB) | **24.201** | **24.784** | **25.231** | **26.464** | **20.982** | **21.203** | **21.771** | **22.364** |
| CT PSNR(dB) | 24.014 | 24.606 | 24.932 | 25.169 | 20.690 | 21.194 | 21.438 | 21.777 |
| WT SSIM | **0.6002** | **0.6529** | **0.6812** | **0.7061** | **0.4197** | **0.4884** | **0.5103** | **0.5384** |
| CT SSIM | 0.5859 | 0.6238 | 0.6516 | 0.6718 | 0.4066 | 0.4632 | 0.4951 | 0.5188 |
| image | Peppers | | | | Camera | | | |
| Rate(BPP) | 0.1500 | 0.2000 | 0.2500 | 0.3000 | 0.1500 | 0.2000 | 0.2500 | 0.3000 |
| WT PSNR(dB) | **27.541** | **28.203** | **29.461** | **30.521** | **27.743** | **28.641** | **29.354** | **31.231** |
| CT PSNR(dB) | 27.101 | 27.861 | 28.261 | 28.488 | 27.001 | 27.730 | 28.519 | 28.867 |
| WT SSIM | **0.7539** | **0.7847** | **0.7948** | **0.8411** | **0.7923** | **0.8179** | **0.8388** | **0.8515** |
| CT SSIM | 0.7354 | 0.7594 | 0.7703 | 0.7816 | 0.7653 | 0.7995 | 0.8236 | 0.8398 |

coding method: Butterfly, Barbara, Zelda, Goldhill, Man, Mandrill, Peppers, and Camera. Figs 12 and 13 show an objective comparison of the performances of the two algorithms. Fig 14 shows some subjective visual comparisons between wavelet SPIHT coding performance and the contourlet-HMM–PCNN SPIHT encoding performances.

As can be seen from the above figures and tables, as far as PSNR and SSIM are concerned, the performance of the contourlet-HMM–PCNN SPIHT coding scheme is better than the simple combination of SPIHT and contourlet. The proposed algorithm resulted in a PSNR of 0.1 to approximately 1.1 dB higher and an SSIM of 0.01 to approximately 0.04 higher than the original algorithm, indicating that the contourlet-HMM–PCNN model actually optimizes the coding process.

In some other comparisons, the compression rates in the contourlet-HMM–PCNN SPIHT and the wavelet SPIHT were both 0.15 bpp, but it could be seen that the wavelet SPIHT still performed better than the SPIHT contourlet with the HMM–PCNN model. Image reconstruction performed by the wavelet SPIHT tends to have clearer boundary areas, especially in the high-frequency parts. Therefore, once the redundancy caused by the LP in the contourlet transform is removed, better results can be obtained.

Similarly, in [48], the author proposed a novel generalized SPIHT algorithm, called set partitioning coding system (SPACS), which has good performance. Therefore, in Table 3, we make a brief comparison between our method and that of SPACS. The image 'barbara' is used and PSNR values are compared at different bit rates. It can be seen from the table that the performance of the contourlet HMM-PCNN based model is poor, which is mainly due to the

**Table 2. Image compression results of SPIHT based on the contourlet-HMM–PCNN model on different bit rate (bpp).**

| Image | Groups of big coefficients | Group of small coefficients | PSNR(dB) | SSIM |
|---|---|---|---|---|
| Butterfly | 0.1500 | 0.1500 | 26.5959 | 0.7231 |
| | 0.2000 | 0.2000 | 26.9279 | 0.7369 |
| | 0.2500 | 0.2500 | 27.0572 | 0.7474 |
| | 0.3000 | 0.3000 | 27.5864 | 0.7658 |
| Barbara | 0.1500 | 0.1500 | 23.5534 | 0.6374 |
| | 0.2000 | 0.2000 | 23.9111 | 0.6699 |
| | 0.2500 | 0.2500 | 24.2305 | 0.6867 |
| | 0.3000 | 0.3000 | 24.6280 | 0.7132 |
| Zelda | 0.1500 | 0.1500 | 31.4631 | 0.8386 |
| | 0.2000 | 0.2000 | 32.0071 | 0.8506 |
| | 0.2500 | 0.2500 | 32.0854 | 0.8667 |
| | 0.3000 | 0.3000 | 32.4087 | 0.8639 |
| Goldhill | 0.1500 | 0.1500 | 26.3568 | 0.6462 |
| | 0.2000 | 0.2000 | 26.9466 | 0.6778 |
| | 0.2500 | 0.2500 | 26.2877 | 0.7034 |
| | 0.3000 | 0.3000 | 27.4705 | 0.7196 |
| Man | 0.1500 | 0.1500 | 24.5138 | 0.6298 |
| | 0.2000 | 0.2000 | 24.9985 | 0.6628 |
| | 0.2500 | 0.2500 | 25.5123 | 0.6900 |
| | 0.3000 | 0.3000 | 25.7233 | 0.7097 |
| Mandrill | 0.1500 | 0.1500 | 21.1476 | 0.4572 |
| | 0.2000 | 0.2000 | 21.5482 | 0.5013 |
| | 0.2500 | 0.2500 | 21.8008 | 0.5303 |
| | 0.3000 | 0.3000 | 22.0296 | 0.5618 |
| Peppers | 0.1500 | 0.1500 | 27.7252 | 0.7569 |
| | 0.2000 | 0.2000 | 28.1414 | 0.7732 |
| | 0.2500 | 0.2500 | 28.4153 | 0.7872 |
| | 0.3000 | 0.3000 | 28.7465 | 0.7920 |
| Camera | 0.1500 | 0.1500 | 27.6717 | 0.7998 |
| | 0.2000 | 0.2000 | 28.1938 | 0.8283 |
| | 0.2500 | 0.2500 | 28.6629 | 0.8472 |
| | 0.3000 | 0.3000 | 29.1115 | 0.8628 |

redundancy of the contourlet transform. However, as shown in other experiments, the contourlet HMM-PCNN model has proven to show better performance than the contourlet model.

We also compared the proposed method with several recently proposed learning based mthods including the method by Theis [49], Balle [50], the fully convolutional vector quantization network(VQNet) [51], the soft-to-hard VQ based method (SHVQ) [52], and a state-of-the art BPG method was also compared [51]. Both the PSNR and the SSIM were used to evaluate the performance of the test methods, and the test images are from Kodak. The test results are shown in Figs 15 and 16. Note that the test image Butterfly is from reference [53].

We mainly tested the algorithms with low bit rate since contourlet transform is redundant and performs poorly at high bit rate encoding. According to the PSNR results, it can be observed that at lower bit rate our model performs similarly with the methods of Theis and Balle. With the bit rate becoming larger, our method is inferior to the other methods.

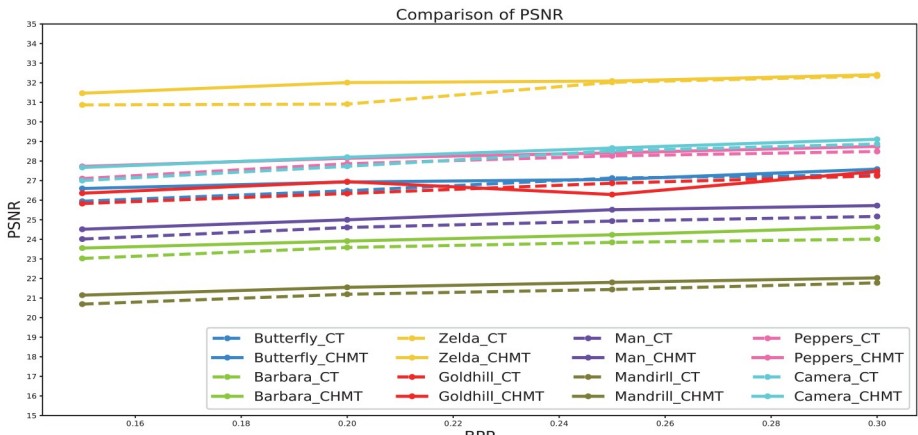

**Fig 12. Comparison of PSNR between the contourlet transform and contourlet-HMM–PCNN algorithms.**

Generally, the BD-rate increases by 6.1% when compared with VQ-Net, reduces by 0.4% when compared with Theis's method, increases by 5.9% when compared with Balle's method. According to the SSIM result, it can be observed that our method performs not so satisfactory when compared to the other methods.

The compression of the classified coefficients with different proportions is as follows: As shown in the flowchart, another novel characteristic of our method is that the original coefficients are separated into two categories. Consequently, we can encode the two parts at different levels to achieve flexible data transmission. As mentioned earlier, the coefficients modelled by HMM represent two states: plain area as low frequency and edges or contours as high frequency. Although in previous experiments, we used the same compression rate for both parts, the fact is that we can use different proportions for the two sets of coefficients. Table 3 shows how different proportions of data affect compression performance, where loop 1 and loop 2 indicate the encoding loops for the two categories, respectively. Generally, with more encoding

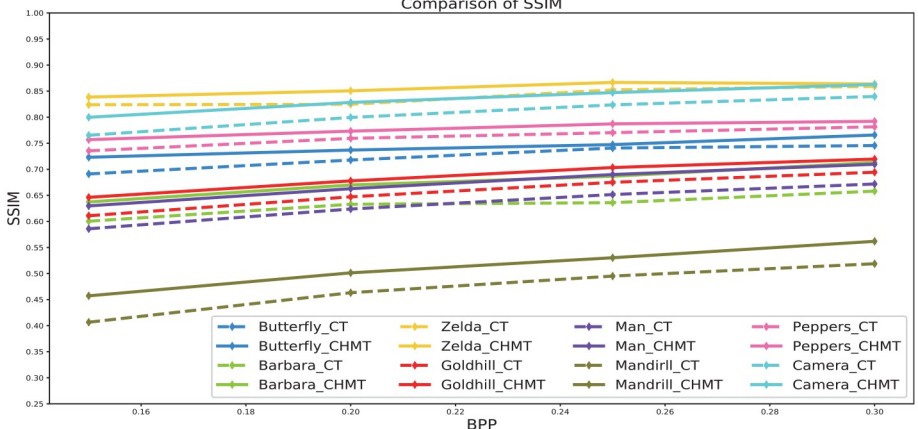

**Fig 13. Comparison of SSIM between the contourlet transform and contourlet-HMM–PCNN algorithms.**

|  | Oringinal | Wavelet SPIHT | Proposed Method |
|---|---|---|---|

**Fig 14. Visual comparisons between contourlet-HMM–PCNN SPIHT and wavelet SPIHT coding performances.**
Reprinted from [40] and [53] under a CC BY license, with permission from IEEE publisher, original copyright 2019, OSA publisher, original copyright 2014, respectively.

loops, the compression performance will be better. Fig 17 offers an intuitive visual evaluation for part of the data listed in Table 4.

In the SPIHT algorithm, there is an idea that important information should be transmitted first, where the value of a certain contourlet coefficient indicates its importance. On the contrary, the state probability produced by the learning procedure is also taken into consideration in our method, which makes it a two-stage selection: the area with saliency is first selected and then the significant coefficients are transmitted in the area. In reality, the first stage can be

**Table 3. Comparison with SPACS algorithm.**

| Bitrate(BPP) | 0.125 | 0.25 | 0.5 | 1 |
|---|---|---|---|---|
| PSNR of ours(dB) | 23.5334 | 24.2305 | 27.8603 | 31.1103 |
| PSNR of SPACS(dB) | 24.7584 | 27.4278 | 31.2271 | 36.1186 |

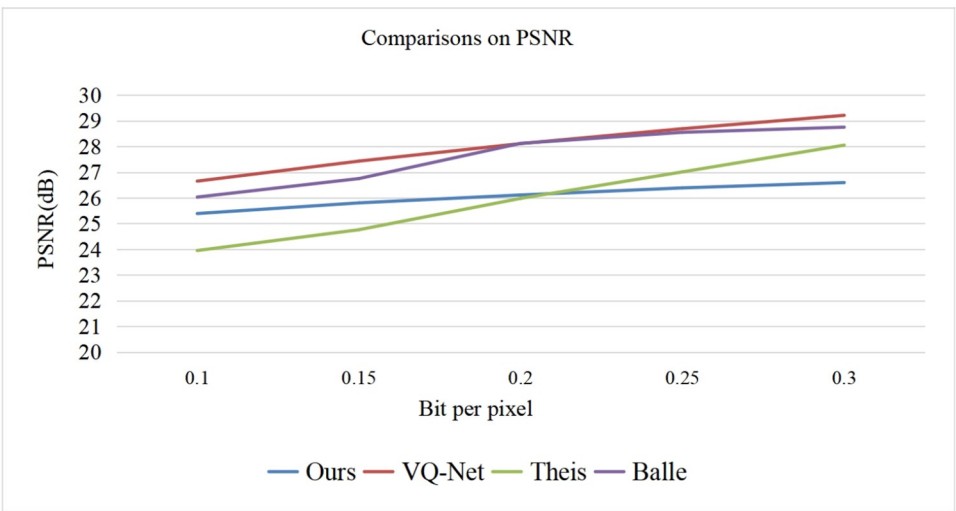

**Fig 15. Comparisons on PSNR.**

considered as saliency detection, which is based on a probabilistic model. The initial coeffi-
cients are classified into the different parts of an image, which is driven by the distribution of
the data itself rather than human-like attention. In the HMM model, there exist only two
states, so the image can only be classified into two parts, and the PCNN is adopted to generate
a pulse output that can only present two states. In our future research, an HMM with more
states will be used to achieve a more refined classification of an image.

## 7 Conclusion

In this paper, we fully utilized both the combined model that took advantage of the modeling
ability of the HMM in spatio–temporal use and the pattern classification ability of a PCNN to
construct a hybrid HMM–PCNN model. Moreover, we modulated the image data to match
the HMM–PCNN model. Finally, we verified the effectiveness of the hybrid HMM–PCNN in
contourlet model part [40] through an image compression application with the well-known
SPIHT algorithm: its performance is better than that of the SPIHT contourlet coding. PCNN

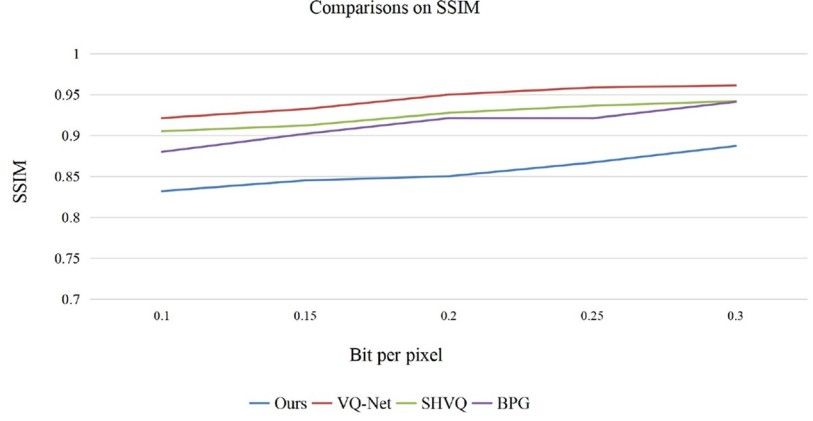

**Fig 16. Comparisons on SSIM.**

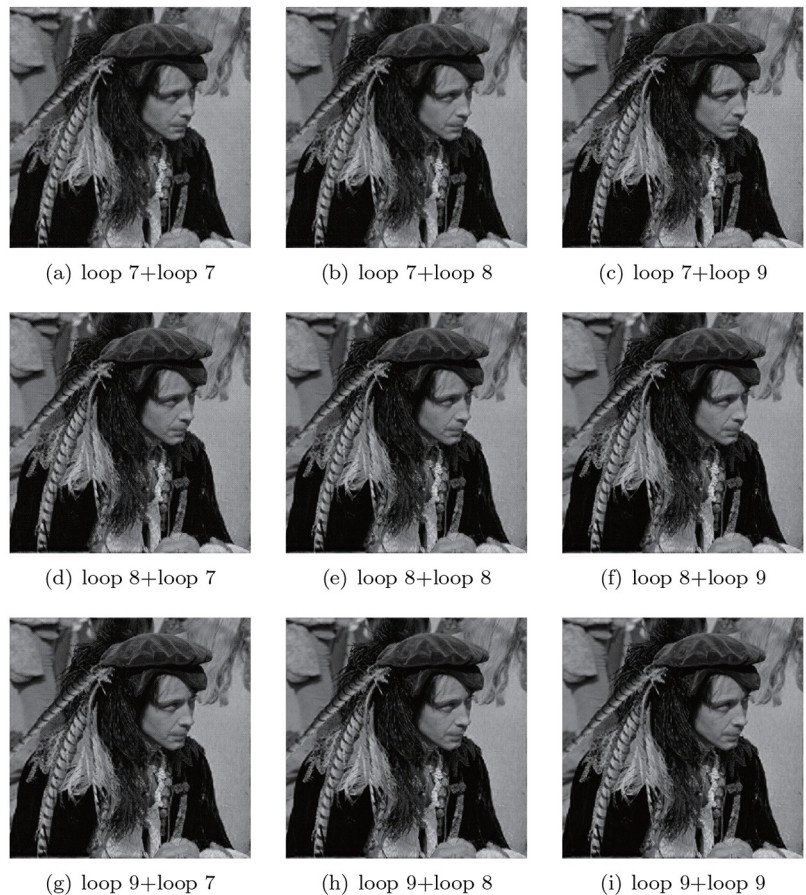

(a) loop 7+loop 7 (b) loop 7+loop 8 (c) loop 7+loop 9

(d) loop 8+loop 7 (e) loop 8+loop 8 (f) loop 8+loop 9

(g) loop 9+loop 7 (h) loop 9+loop 8 (i) loop 9+loop 9

**Fig 17. Visual performance of compression with different proportion of encoding level, for the both two kinds of coefficients, deeper encoding loop results in better reconstruction.**

has been proven to be a good classifier in recognizing the features captured by the HMM model, resulting in a more efficient coding method. In future research, we will further use human visual attention and saliency features to create more effective sparse representations of texture images.

**Table 4. Compression with different proportion.**

| Loop1 | Loop2 | PSNR(dB) | SSIM |
| --- | --- | --- | --- |
| 6 | 7 | 18.5784 | 0.2098 |
| 6 | 8 | 19.1519 | 0.2342 |
| 6 | 9 | 19.6488 | 0.2624 |
| 7 | 7 | 22.0211 | 0.3702 |
| 7 | 8 | 23.1609 | 0.3711 |
| 7 | 9 | 23.7102 | 0.5002 |
| 8 | 7 | 24.1670 | 0.5184 |
| 8 | 8 | 25.6844 | 0.6302 |
| 8 | 9 | 26.5936 | 0.7080 |
| 9 | 7 | 25.6114 | 0.6288 |
| 9 | 8 | 26.9780 | 0.7417 |
| 9 | 9 | 27.3179 | 0.7892 |

## Supporting information

**S1 Text.**
(TXT)

**S1 Code.**
(RAR)

**S1 Data.**
(XLSX)

## Author Contributions

**Investigation:** Junjie Yang.

**Methodology:** Guoan Yang, Zhengzhi Lu.

**Software:** Junjie Yang.

**Writing – review & editing:** Yuhao Wang.

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
