## [Decision Letter · Decision Letter 0]

8 Apr 2020

PONE-D-19-32659

A combined HMM–PCNN model in the contourlet domain for image data compression

PLOS ONE

Dear Professor Yang,

Thank you for submitting your manuscript to PLOS ONE. After careful consideration, we feel that it has merit but does not fully meet PLOS ONE’s publication criteria as it currently stands. Therefore, we invite you to submit a revised version of the manuscript that addresses the points raised during the review process.

We would appreciate receiving your revised manuscript by May 23 2020 11:59PM. To enhance the reproducibility of your results, we recommend that if applicable you deposit your laboratory protocols in protocols.io, where a protocol can be assigned its own identifier (DOI) such that it can be cited independently in the future. For instructions see: http://journals.plos.org/plosone/s/submission-guidelines#loc-laboratory-protocols

We look forward to receiving your revised manuscript.

Kind regards,

Zhaoqing Pan, Ph.D.

Academic Editor

PLOS ONE

2.  We note that Figures 9-11, 14-15 in your submission contain copyrighted images. All PLOS content is published under the Creative Commons Attribution License (CC BY 4.0), which means that the manuscript, images, and Supporting Information files will be freely available online, and any third party is permitted to access, download, copy, distribute, and use these materials in any way, even commercially, with proper attribution. For more information, see our copyright guidelines: http://journals.plos.org/plosone/s/licenses-and-copyright.

1.    You may seek permission from the original copyright holder of Figures 9-11, 14-15 to publish the content specifically under the CC BY 4.0 license.

Reviewers' comments:

Reviewer's Responses to Questions

**Comments to the Author**

1. Is the manuscript technically sound, and do the data support the conclusions?

Reviewer #1: Yes

Reviewer #2: Yes

2. Has the statistical analysis been performed appropriately and rigorously? 

Reviewer #1: Yes

Reviewer #2: Yes

3. Have the authors made all data underlying the findings in their manuscript fully available?

Reviewer #1: No

Reviewer #2: Yes

4. Is the manuscript presented in an intelligible fashion and written in standard English?

Reviewer #1: Yes

Reviewer #2: Yes

5. Review Comments to the Author

Reviewer #1: Question1:

The motivation of the proposed combined HMM–PCNN model should be described more clearly and consistently, and it should be reflected in the method.

Question2:

There are too many experimental results in Tables 1 and Table 2, and it looks a bit confusing to analyze the experiment results. Maybe authors can adjust the table organization or bold the important results.

Question3:

The experiments should enrich the comparison between the proposed method and other state-of-the-art algorithms further. Their method is just compared with one method presented in 2016. The reviewer hopes to see more discussions and experiments to demonstrate the efficiency and robustness of the proposed method.

Reviewer #2: The author should spend more descriptions on the method they proposed. It is confused to introduce too many methods used before. In the meanwhile,although the author focus on a different transform for image compression, there is still lack of comparison with SOTA learned compression methods.

1）More intuitive data should be provided such as BD-rate reduction based on a specific anchor.

2) Common test results on well-known data Kodak should be provided.

6. PLOS authors have the option to publish the peer review history of their article (what does this mean?). If published, this will include your full peer review and any attached files.

Reviewer #1: No

Reviewer #2: No

---

## [Author Response · Author response to Decision Letter 0]

1 May 2020

Please see the response to the editor's comments and the response to the reviewers.

---

## [Editor Report · Decision Letter 1]

30 Jun 2020

A combined HMM–PCNN model in the contourlet domain for image data compression

PONE-D-19-32659R1

Dear Dr. Yang,

We’re pleased to inform you that your manuscript has been judged scientifically suitable for publication and will be formally accepted for publication once it meets all outstanding technical requirements.

Kind regards,

Zhaoqing Pan, Ph.D.

Academic Editor

PLOS ONE

Additional Editor Comments (optional):

All questions have been address, and can be accepted for publication.

---

## [Editor Report · Acceptance letter]

13 Jul 2020

PONE-D-19-32659R1 

A combined HMM–PCNN model in the contourlet domain for image data compression 

Dear Dr. Yang:

I'm pleased to inform you that your manuscript has been deemed suitable for publication in PLOS ONE. Congratulations! Your manuscript is now with our production department. 

Kind regards, 

on behalf of

Dr. Zhaoqing Pan 

Academic Editor

PLOS ONE